# Care Recommendations for the Chronic Risk of COVID-19: Nursing Intervention for Behaviour Changes

**DOI:** 10.3390/ijerph19148532

**Published:** 2022-07-12

**Authors:** Alexandra González Aguña, Marta Fernández Batalla, Blanca Gonzalo de Diego, María Lourdes Jiménez Rodríguez, María Lourdes Martínez Muñoz, José María Santamaría García

**Affiliations:** 1Henares University Hospital, Community of Madrid Health Service (SERMAS), 28822 Madrid, Spain; 2Research Group MISKC, Department of Computer Science, University of Alcala, Polytechnic Building, University Campus, Barcelona Road Km. 33.6, 28805 Alcalá de Henares, Spain; marta.fdezbatalla@gmail.com (M.F.B.); blanca94gd@gmail.com (B.G.d.D.); lou.jimenez@uah.es (M.L.J.R.); chesantgar@hotmail.com (J.M.S.G.); 3Torres de la Alameda Health Centre, Community of Madrid Health Service (SERMAS), 28813 Madrid, Spain; 4Meco Health Centre, Community of Madrid Health Service (SERMAS), 28880 Madrid, Spain; 5Computer Science Department, University of Alcala, 28805 Madrid, Spain; 6Care Management, Community of Madrid Health Service (SERMAS), 28020 Madrid, Spain; mlourdes.martinez@salud.madrid.org

**Keywords:** community health nursing, COVID-19, counselling, nursing care

## Abstract

The COVID-19 pandemic is a challenge for health systems. The absence of prior evidence makes it difficult to disseminate consensual care recommendations. However, lifestyle adaptation is key to controlling the pandemic. In light of this, nursing has its own model and language that allow these recommendations to be combined from global and person-centred perspectives. The purpose of the study is to design a population-oriented care recommendation guide for COVID-19. The methodology uses a group of experts who provide classified recommendations according to Gordon’s functional patterns, after which a technical team unifies them and returns them for validation through the content validity index (CVI). The experts send 1178 records representing 624 recommendations, which are unified into 258. In total, 246 recommendations (95.35%) are validated, 170 (65.89%) obtain high validation with CVI > 0.80, and 12 (4.65%) are not validated by CVI < 0.50. The mean CVI per pattern is 0.84 (0.70–0.93). These recommendations provide a general framework from a nursing care perspective. Each professional can use this guide to adapt the recommendations to each individual or community and thus measure the health impact. In the future, this guideline could be updated as more evidence becomes available.

## 1. Introduction

The coronavirus disease, COVID-19, emerged in December 2019 in China and managed to spread until three months later, when it was classified as a pandemic by the World Health Organization (WHO). During the first months, health systems had to navigate a highly contagious, global illness with high morbidity and mortality while having no prior evidence, leading to a restructuring of the available resources [1,2,3].

The first health measures were based on the modification of individual and social behaviour to prevent transmission [3,4,5]. Subsequently, the development of vaccines was implemented on a large scale and showed positive results [6,7]. However, two years after the first cases were reported, lifestyle measures remain essential, and COVID-19 has not disappeared [8]. At present, this infection shows an epidemiological pattern of waves with temporary periods of increased infections and hospitalized people. The risk of contagion from the virus SARS-CoV-2 (which causes COVID-19) has become chronic, and the population of any age faces a coexistence with the virus [9].

Given this situation, nursing has proven to be the backbone of any health system and the front line for the prevention and treatment of disease [10].

In 2020, within the framework of the International Year of the Nurse and the Midwife, the WHO and the International Nursing Council published a report on the situation of nursing in the world. This report shows the differences between countries regarding training programs, competencies, professional skills, and access to health policies and management positions [11]. However, the nursing profession still needs to improve the evidence of the impacts of its clinical interventions on population health outcomes [12]. Some actions are based on the modification of behaviours (lifestyles), focused on individuals, families, and communities and largely directed towards the promotion of health and disease prevention, beyond the treatment of illness and recovery of health.

Nursing has its own object of knowledge, *care*, which has developed theoretical models and even has standardized languages [13,14,15,16]. The metaparadigm of care provides the framework when interesting phenomenon elements of nursing are proclaimed as nursing theories [13]. The metaparadigmatic elements of care include the person, the environment, and health [13].

The study of these elements to aggregate the knowledge and management of care led to the development of a theoretical model called the Knowledge Model about Person Care [17,18].

This model emerged in the interdisciplinary field of nursing and computer science from doctoral studies and is applied in the clinical setting [19,20]. The Knowledge Model about Person Care defines the elements of the metaparadigm of care and establishes the meanings of the relationships between them.

In this model, care is a continuum, ever present in people’s lives whether professionally or not and beyond clinical treatment [18]. The person is the central element of care, and it is defined from their *vulnerability*. Vulnerability is an essential human condition; it accompanies a person throughout life and is linked to the need and competence for self-care [21,22]. The environment circumscribes the person, constituting an open, complex, and multidimensional system in constant interaction that defines the *risk* in care. The environment can be divided into interpersonal elements (family, parental, social) and geopolitical elements (physical environment, resources, health system) [13,23]. The union of vulnerability and risk gives rise to *predisposition* [18,19,20].

Furthermore, the person may present care limitations: lack of knowledge, ability, or attitude. The *predisposition* together with these limitations to care gives rise to the *potentiality* for care problems. Problems may be in the *subclinical phase* (no signs or symptoms are evident) or *clinical phase* (the problem is expressed in signs and symptoms). The final degree reached by this process is defined as *severity*. Thus, these care problems determine the level of health of people, and the consequences of these problems (physical, mental or social) will condition the care of the person in the future [18,19,20].

Health arises as the result of caring for the person in their environment at every moment and, in addition, as a resource for the continuum of care [13,24]. Nursing is the discipline of care and, therefore, the profession responsible for leading the care of the population.

Figure 1 shows the conceptual summary of the Knowledge Model about Person Care.

This model is centred on the person, designed from the individual person’s perspective as well as those of their families and communities, all of whom benefit from health services. Caring attention focuses on needs with an integrated approach. The environment frames the situation where care is performed, such as home, school, work, or health centre, always respecting the sociocultural mandate to provide culturally consistent care. Caring interactions are continuous in people’s lives, not only when there is an illness [25,26,27].

Nurses have a coaching role of helping people develop their highest competence and become the main characters in their care management. Nurse intervention that acts for the person will be limited to timely moments [27].

The Knowledge Model about Person Caretakes Orem’s theory as a reference for the relationships between need, care competence, and nursing support systems. Orem’s *Self-Care Deficit Theory* points out that nurses act when there is a self-care deficit (the demands for care exceed the competencies of the individual and even the competency provided by family care). These self-care alterations can occur during life stage transitions and through health diversion processes such as COVID-19. Nurses address these situations through the nursing care process, where they establish the clinical judgment that is expressed through nursing diagnoses [13,14,15,28,29]. Professional intervention includes everything from full compensation systems in cases of critical severity to support and education actions based on the use of recommendations [13].

Recommendations are “*statements designed to help end-users make informed decisions on whether, when and how to undertake specific actions such as clinical interventions, diagnostic tests or public health measures, with the aim of achieving the best possible individual or collective health outcomes*” [30]. These recommendations reach to the population as health advice as brief, opportunistic (taking advantage of professional meetings) interventions that include information based on evidence and that motivate behaviour change [31].

At present, health lifestyle recommendations are the main axis of sanitary action against COVID-19, although there are developed vaccines for this health problem [8].

### Main Aim

The aim of this study was to design a guide for care recommendations against chronic risk of COVID-19. The guide was designed from a person-centred model. The person who is using a health care system is the main character in their *vital care continuum* and who needed to be trained in decision-making to modify and adopt health care behaviours.

## 2. Materials and Methods

### 2.1. Design

The design was based on a mixed-method study for the development and validation of a guide for care recommendations. The guide was elaborated through a person-centred design and with a perspective based on a nursing care model.

The research team explained to the participants the Knowledge Model about Person Care. Person-centred care suggests a needs analysis from the individual person perspective, incorporating a person’s family; community; cultural preferences; and home, school, and work environments. People are the main characters in their care management and therefore of their behaviour modification

The study methods were compliant with the Good Reporting of a Mixed Methods Study (GRAMMS) checklist [32].

### 2.2. Procedure

The procedure combined two phases because this mixed method allowed for (a) collecting the available published knowledge and (b) obtaining feedback from experienced experts regarding this field in which previous evidence is very scarce. After the data collection, phase II was to validate each expert recommendation by applying a consensus index. This procedure combined different techniques that reduced the limitations that appear when only one method is used.

#### 2.2.1. Phase I—Acquisition of Expert Knowledge

The first phase of the study was carried out from May 2020 to June 2021 and entailed knowledge acquisition through elicitation with a group of experts.

The study began from the Care Management of Madrid Health Service (SERMAS), which formed a coordinating team of four experts. The coordinating group included three women and one man, with a mean age of 44.75 (33–63) years, all with more than ten years of professional experience; three held doctorate degrees, and one woman held a master’s degree. All of them had teaching and research experience. The four experts represented the nursing profiles of management, teaching, and care in hospitals and health centres.

The data collection for the elicitation of knowledge used the Microsoft Excel^®^ tool and was based on some files designed by the coordinating team.

The files contain some tables that take as a reference the eleven functional patterns of Marjory Gordon [33] because they offer a complete and integral structure of the assessment of care of the person in their current context and also because it is the system used for registration in the electronic clinical records in primary care throughout the region. All the patterns were broken down into elements to guide the participants in the different aspects that each pattern integrated.

The table proposal for each functional pattern was approved by piloting with a group of five clinicians external to the research. The objective of the pilot study was to ensure that each form could be completed in a self-administered way. These experts were nursing professionals with care dedication and previous experience in research as participants in groups of experts. These professionals provided recommendations to improve the clarity of the content and the explanation that should be sent to the participants aimed at achieving individual and independent fulfilment without third party intervention.

Each functional pattern presented the data structure shown in Table 1. 

All files were sent to a group of clinical experts recruited through intentional sampling representing health centres throughout the region.

The group included six participants, all women, five nurses and a physiotherapist, all with more than ten years of professional experience. Nurses represented all levels of care: two at an adult hospitals, one at a children’s hospital, one in community and family primary care, and one in an extra-hospital emergency service. Four of them had postgraduate degree (master or speciality), and three of them had teaching experience.

The participants included nurses from hospitals because they participated in the treatment of diseases but also in prevention and promotion. Hospital professionals attended to the demands for health care in the emergency department in cases of suspicion and diagnoses of COVID-19 and resolved doubts from relatives of hospitalized people; in addition, they received training for subsequent mass vaccination campaigns for the population, which included care advice. The physiotherapist was included because in Spain, this discipline is related to nursing direction in many health centres and works together with nursing to rehabilitate patients at hospitals.

All participants had clinical experience in health care for people with COVID-19. All had previously participated in at least one care research project.

No one of the participants represent a scientific society or a company related to health or pharmaceuticals that could have created any conflicts of interest that could have affected the results.

The coordinating group held a face-to-face session lasting an hour and a half to present the research project to the group. The session was divided into one hour for an exposition of the study aim, methodology, duration, and intended results and thirty minutes for answering questions and doubts. All participants were informed and agreed to participate in the project.

Once participation was accepted, the coordinating group sent the participants an email with the summarized information (participants’ rights, aim, methodology, estimated time, findings diffusion), and then the patterns were sent with a time interval of one month between each shipment. Each participant received an Excel file with the table corresponding to the functional pattern and had to complete it within one month, maximum two months. In this first phase, each participant answered individually and anonymously and justified each recommendation with at least one bibliographic source (clinical guides, governmental web, published research). The participant returned the completed file to the coordinating group via email.

At the end of the collection phase, the coordinating team grouped all the contributions of each functional pattern in the same file and unified the recommendations based on their content. The contributions with the same main focus (equivalence by word or meaning) were unified in a single recommendation to which all the bibliographical references provided by the experts were linked. These files were used in the following study phase.

#### 2.2.2. Phase II—Validation of Recommendations

The second phase of the study was carried out from July 2021 to December 2021 and applied expert group consensus methodology. The coordinating group returned the result of the previous unification to the group of experts.

Each participant received an Excel file to complete a self-administered questionnaire. The questionnaire consisted of a table with the unified recommendations of each pattern together with the percentage of participants who had provided some record related to it.

The question that guided the assessment was “*Do you consider this recommendation adequate for the care of people at risk or diagnosed with COVID-19?*”. The evaluation used a 5-point Likert scale, where 1 means “not representative” and 5 means “fully representative”. In addition, each recommendation included a section so that the experts could add comments in free text on aspects such as the reason for their score or suggestions for changes in the wording. This 5-point rating scale was the same as that for nursing outcomes classification (NOC) used in previous similar studies [34].

Each pattern was sent every two weeks. The experts had that time to carry out the evaluation and return it to the coordinating team via email.

The analysis of the responses was based on the content validation index (CVI). The CVI is the result of dividing the number of positive responses (4 or 5 points on the Likert scale) by the total number of responses. Previous research references established the cut in different ratings [34,35,36,37]. For this study, the interpretation of the result was divided into three categories: <0.50 = “Not approved”, 0.50–0.80 = “Approved”, and >0.80 = “Approved with high validation”. For the high validation of an item, at least five participants had to give the item the highest ratings.

The result of the scoring was returned to the group of participants as final feedback on the agreed proposal. The participants validated the recommendations as a unique group, and each of them had the opportunity to share their degree of agreement with all of the recommendations agreed on in the first phase.

Data were next acquired and validated by the same group of experts: The first phase was individual, anonymous, and based on publications, and the second phase was collective, by consensus, and adapted to the clinical realities the professionals had observed.

## 3. Results

### 3.1. Phase I—Acquisition of Expert Knowledge

The experts sent a total of 1178 records on potential recommendations for the 11 functional patterns. All 54 elements into which the functional patterns were broken down included at least 1 record. These records constituted 624 contributions, which were later unified into 258 recommendations.

The summary of the distribution of recommendations for each pattern can be seen in Table 2. The functional patterns with the most recommendations were cognition–perception and roles–relationships. At the opposite extreme, the functional pattern with the least representative weight was elimination.

### 3.2. Phase II—Validation of Recommendations

In total, 246 recommendations (95.35%) were validated, with a mean CVI by patterns of 0.84. The pattern with the lowest score was sexuality–reproduction, with CVI = 0.70, and the pattern with the highest score was coping–stress tolerance with CVI = 0.93.

The functional pattern self-perception–self-concept showed the highest validation at 83.33%, and sexuality–reproduction showed the lowest percentage with 28.57%. Conversely, this same functional pattern on sexuality obtained the highest percentage of non-validated recommendations, 19.05%.

Some participants did not assess some recommendations because they considered that they were not able to issue an assessment, either because the content of the recommendation was foreign to their discipline, in the case of the physiotherapist, or because they lacked specific clinical experience with the area, in the case of the nurses. In total, there were 30 blank responses (22.41%).

Detailed information for each functional pattern is shown in Table 3.

The recommendations with high validation were 170 (65.89%).

In addition, 23 recommendations (8.91%) achieved a CVI = 1.00, either the maximum score (all participants awarded 5 points) or the maximum score minus one point. Behavioural areas such as diet, exercise, positive mental health and coping were the recommendations that most participants identified when providing references.

The recommendations with the highest validations are shown in Table 4.

On the opposite side, 12 recommendations (4.65%) were not validated because they obtained a CVI < 0.50. All these recommendations presented a percentage of references of 17%, that is, only one of the six experts had provided some source to substantiate it.

The recommendations with the lowest CVI were the suspension of mass vaccination campaigns, the delay in changing the intrauterine device for one more year than recommended, the marking of personal belongings, and the idea that older people do not express uncertainty about the future.

These recommendations were not validated for various reasons. The lack of minimal evidence or sufficient study is alleged for the recommendations of electrocardiographic monitoring, vitamin supplements, increasing the recommended time of the intrauterine device, or assuming that retired people do not show uncertainty. In addition, the fear of contagion or boredom is applicable to any vital stage. For other recommendations, such as menstrual hygiene, prevention of sexually transmitted diseases, or assisted reproduction, the experts did not consider that these were specifically related to COVID-19 or that their consequences were due to confinement. Regarding the use of active videogames, two experts stated that they consider it generic and not specific and recommended it be included as a technique or resource within broader recommendations, such as multicomponent programs. For vitamin D supplementation and the development of a new body image, the experts did not consider that there was sufficient evidence; they only appeared in the literature because they were being or could be the subject of study.

Finally, some recommendations explicitly related to the clinical treatment of COVID-19 and its consequences in professional health care should be highlighted. Cutaneous drug reactions were being studied but did not allow for a clear and specific recommendation. Mask regulations were constantly changing, and, massive vaccination campaigns were maintained in the region, with special attention to maintaining the childhood vaccination schedule.

The recommendations with less validation are collected in Table 5.

## 4. Discussion

The aim of this study was to develop a recommendation guide for people who were at risk of or diagnosed with COVID-19. A guide that, unlike the publications prior to the study, was based on a nursing care model called the Knowledge Model about Person Care, centred on the person in any context and providing care from a comprehensive, integrated, and continuous perspective.

The results show care recommendations for all functional patterns, covering all life stages and even specific circumstances such as pregnancy. Likewise, these recommendations can be transferred to different contexts due to their concrete and brief form that allows for their adaptation to individuals or groups.

However, despite the importance of the recommendations in the prevention of the pandemic, research studies on the use and effectiveness of this intervention are scarce. The main sources of recommendations for the general population are websites such as the WHO, a global reference entity in this area, and government websites, such as the ministries of health of the different countries [38,39].

At the end of the study, no publications were found with a similar objective to the one proposed in this research or with an approach from nursing care models.

The results obtained can be linked to each of the elements of the nursing care model shown in the introduction.

Regarding the person as the centre of interest, several aspects can be highlighted.

The person is an individual who can be characterized according to their age, vital processes, and previous health alterations or, as highlighted in this study, by their employment status.

Age differentiates two vulnerable population groups at the extremes. These two groups were already identified in the first WHO clinical guidelines and have been maintained and developed in the latest version of their clinical management guide [4,8]. Childhood suffered changes to their daily lives because school was interrupted, and time at home produced a risk of increased hours in front of screens (television, video games). In addition, periods of isolation in this group pose a greater risk of suffering fear or anxiety problems [40]. Elderly people present a similar risk due to isolation, and it is recommended that contact be maintained with those who live alone, that multicomponent exercise be encouraged, and that confusion be considered a possible sign of infection [41]. It should be noted that the experts did not validate the proposal that assumes that retired people do not show uncertainty about the future.

Regarding vital processes, the recommendations include pregnancy, breastfeeding, and bereavement. Pregnancy and lactation are a period of special interest due to the women’s changed needs and demands and their impacts on their foetuses or new-borns and also because studies on pharmacological treatments in this population are scarcer than studies on typical adult populations. Another situation of special interest in the pandemic is grief. High mortality and confinement conditions impacted ways of grieving the losses of loved ones [42].

On the other hand, previous health alterations have been the subject of interest. Lopez et al. [43] provide a set of recommendations for renal transplant patients. These recommendations show, as in this study, that care goes beyond the origin of the deviation from health and reaches general measures on social distance, hygiene, nutrition, or adherence to the therapeutic plan through teleconsultation. Regarding feeding, Muscogiuri et al. [44] summarize the main characteristics of an adequate diet during the confinement period for the general population. This diet includes the right proportions of macronutrients and micronutrients, highlighting vitamins C and D and zinc for the immune system. In this study, the experts did not approve the recommendation of vitamin D supplements because they considered that the Mediterranean diet and the possibility of sunlight in this region of Spain were sufficient. This recommendation needs to be reviewed in other countries with different diets and sunlight.

Finally, within health deviations, there are also studies that provide recommendations from groups of experts based on the available evidence and focused on specific circumstances. Shanthanna et al. [45] highlight that the pandemic led to and exacerbated negative consequences such as isolation, stigma, and financial stress. Health professionals must ensure continuity of care using available technological resources. Bresadola et al. [46] offer practical recommendations for controlling the risk of infection in the perioperative setting.

These last two publications share a research interest in health professionals as a population at risk for COVID-19 infection. This situation does not derive from any personal health conditions or vulnerabilities but from their exposure as professionals and they conditions they work under.

Accordingly, the recommendations refer to environments of special interest such as health centres, residences, and schools. These environments had specific regulations during lockdown and subsequent reopening. Health centres had to adapt material and human resources by reinforcing services such as intensive care units in hospitals or, later, health centres for mass vaccination. The homebound experienced significant impacts in the first months, with high morbidity and mortality as a result of the fragility of the population. Schools closed during the period of confinement, and teaching strategies had to be modified. In all these environments, technologies helped bridge distances and isolation, allowing contact with health professionals, caregivers, and teachers to be maintained. The results reported in the study coincide in highlighting the importance of technologies [47,48,49,50]. 

### 4.1. Limitations and Future Lines

Some limitations of the study need to be highlighted. First, the present study began when no vaccine had been approved, and therefore, this recommendation was not added until the final revision.

The research uses a methodology that integrates the consensus of experts from recommendations that have at least one referenced bibliographic source. Nevertheless, the low number of publications on recommendations and, more specifically on nursing care recommendations for COVID-19, make the initial level of evidence weak. No clinical research was found on the effectiveness of this type of nursing intervention, and therefore, the level of evidence is reduced to expert consensus and literature reviews, as occurs in this study. In contrast, studies on pharmacological treatments and vaccines are abundant and based on randomized clinical trials. In the future, the identified recommendations can be considered a common basis for the nursing profession to generate specific research that clinically validates the recommendations and increases their level of evidence.

The limited number of experts should also be considered a limitation. Studies with a larger number of participants or focused on certain professional contexts can improve the methodological quality of the research. This study only included professionals who work directly in patient-centred care and who depend organizationally on nursing management. Medical professionals were not included because their disciplinary object is in the disease. However, future studies may include other perspectives such as specialists in family and community medicine.

Finally, research has shown that the context has a special interest in an infectious process because risk control is essential. This study is focused on the individual, but some contextual issues appear at both interpersonal (families, dependent care, specific groups) and geopolitical (health centres, schools) levels. This element of the metaparadigm of care can be studied in greater depth with similar studies.

### 4.2. Relevance to Clinical Practice

The research carried out shows the importance of comprehensive and integrated care of the person in the face of any health alteration. Nurses are the guarantor of this care and must lead clinical decision-making in this field of knowledge. This approach has already been highlighted in the Nursing Now strategy and is on the agenda for the 2030 action plan [51,52].

Likewise, the results of this study offer a common basis for all nursing professionals because it groups all the recommendations under a common disciplinary framework. These recommendations can be transferred to clinical practice through health advice, and future studies can measure the effectiveness of this type of educational intervention in behaviour change.

This care recommendation guide is directed towards application in the regional context of Madrid, although the participants contribute to a general vision from all nursing clinical perspectives and support each recommendation with previous international publications. The guide is limited in its practical application because results have limited external validation. This limitation is justified because of the person-centred perspective used in this study, that is to say, according to the individual characteristics of the target population considering their environments and their cultural preferences. Extrapolating the recommendations to other regions requires an adapted study that considers care realities in each populational group.

## 5. Conclusions

The nursing perspective in approaching any health situation, from healthy person to health deviation, has value for society. Care models allow for transcending the disease or reason for consultation that leads the citizen to demand health care. The care models allow us to open our gaze by placing the person at the centre of the system considering not just the individuals but their social and physical contexts.

These areas are visible in the recommendations because they include aspects explicitly linked to COVID-19 (alteration in breathing, hygiene, and isolation measures) and, in addition, more general aspects that are related to the lifestyle of the person.

Educational interventions that seek to modify behaviour start from understanding the baseline situation of the person and their vulnerability. In this sense, these results offer recommendations applicable to the general population such as diet, exercise, rest, social relations, sexual education, and physical and cognitive activity as well as aspects of coping and spirituality. On the analysis of this situation, the changes produced by the risk situation or diagnosis of COVID-19 exposed care problems that can be labelled into standardized nursing language.

This change of focus in the investigation of health problems differentiates nursing and makes it visible, as well as providing a basis for further studies that could delve into specific areas but would maintain the same disciplinary perspective of care.

## Figures and Tables

**Figure 1 ijerph-19-08532-f001:**
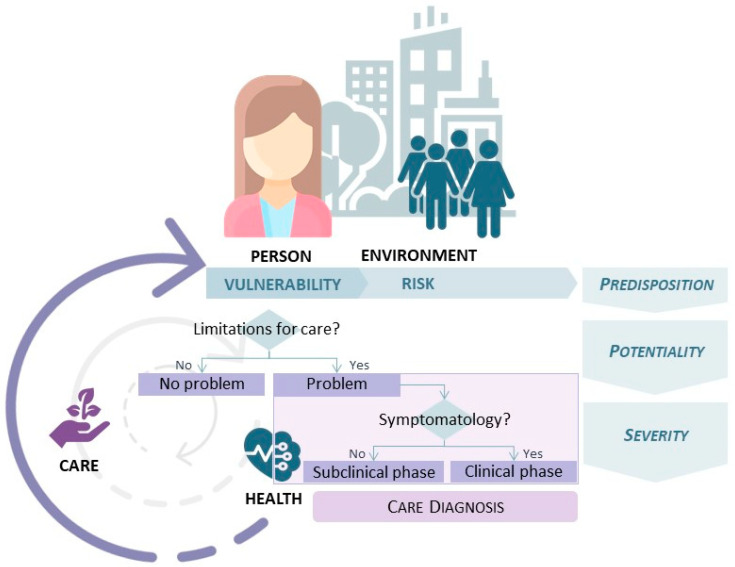
Knowledge Model about Person Care.

**Table 1 ijerph-19-08532-t001:** Data for functional pattern recommendations.

Data	Meaning
Recommendation element	Subunit that guides the global assessment of the different aspects that constitute each pattern
Recommendation regarding	Cell for recording the contributions of each participant that will be converted into a recommendation
Bibliographicl source	Cell for recording the bibliographic sources that support the contribution made.

**Table 2 ijerph-19-08532-t002:** Recommendations by functional pattern.

Functional Pattern	Records Provided (*n*)	Recommendations *n* (%)
1. Health Perception–Health Management	73	30 (11.63)
2. Nutrition–Metabolism	84	27 (10.47)
3. Elimination	26	12 (4.65)
4. Activity–Exercise	74	30 (11.63)
5. Sleep–Rest	33	14 (5.43)
6. Cognition–Perception	58	31 (12.02)
7. Self-Perception–Self-Concept	48	18 (6.98)
8. Roles–Relationships	71	31 (12.02)
9. Sexuality–Reproduction	53	21 (8.14)
10. Coping–Stress Tolerance	53	20 (7.75)
11. Values–Belief	51	24 (9.30)

**Table 3 ijerph-19-08532-t003:** Validation of the recommendations by functional pattern.

Functional Pattern	Mean CVI	Max. CVI	Min. CVI	Recommendations High Validation *n* (%)	Recommendations Not Approved *n* (%)	No Answer *n* (%)
1. Health Perception–Health Management	0.83	1.00	0.20	20 (66.67)	2 (6.67)	6 (3.33)
2. Nutrition–Metabolism	0.87	1.00	0.40	19 (70.37)	1 (3.70)	5 (3.09)
3. Elimination	0.87	1.00	0.40	9 (75.00)	1 (8.33)	4 (5.56)
4. Activity–Exercise	0.82	1.00	0.33	18 (60.00)	1 (3.33)	2 (1.11)
5. Sleep–Rest	0.83	1.00	0.50	9 (64.29)	0 (0.00)	2 (2.38)
6. Cognition–Perception	0.81	1.00	0.50	20 (64.52)	0 (0.00)	5 (2.69)
7. Self-Perception–Self-Concept	0.87	1.00	0.33	15 (83.33)	1 (5.56)	0 (0.00)
8. Roles–Relationships	0.83	1.00	0.33	22 (70.97)	1 (3.23)	2 (1.08)
9. Sexuality–Reproduction	0.70	1.00	0.17	6 (28.57)	4 (19.05)	4 (3.17)
10. Coping–Stress Tolerance	0.93	1.00	0.67	16 (80.00)	0 (0.00)	0 (0.00)
11. Values–Belief	0.87	1.00	0.17	16 (66.67)	1 (4.17)	0 (0.00)

**Table 4 ijerph-19-08532-t004:** The approved recommendations with highest scores.

Functional Pattern	Recommendation	Contributing Participants %	CVI
1	- Know the signs and symptoms of COVID-19 infection.	17%	1.00
	- Use individual protection equipment (PPE) according to current protocol.	17%	1.00
2	- Maintain a balanced and healthy diet, adjusted to the specific conditions of the person.	83%	1.00
3	- If possible, use a separate toilet. In case of shared use and COVID-19 diagnosis, take extreme hygiene measures (hand washing, use bleach for disinfection).	67%	1.00
	- Monitor the appearance of vomiting and diarrhoea; they can be symptoms of COVID-19 infection.	67%	1.00
4	- Avoid physical inactivity; promote exercise adapted to each person and their functional capacity with progression as tolerated.	83%	1.00
	- For the elderly: carry out multicomponent programs that include aerobic, strength, flexibility, and balance exercises.	50%	1.00
	- For people with respiratory sequelae from COVID-19: perform aerobic exercise to recover previous basal capacity and improve psychological condition. Techniques to improve ventilation and drainage of secretions can be applied to promote improvement. Sports activity should be resumed after 7–10 days with mild-moderate intensity.	50%	1.00
5	- Maintain an hour of exposure to daylight and reduce the use of screens (TV, computer, tablet, mobile), especially before bed.	33%	1.00
6	- Promote cognitive training exercises (stimulation of spatial and temporal orientation, memory, perception), especially in situations of isolation.	33%	1.00
7	- Promote positive mental health, with motivation for healthy behaviour, contact with family and friends, and the use of reliable sources of information.	83%	1.00
	- Identify recurring thoughts and emotions, analysing fears.	17%	1.00
	- Identify coping strategies to overcome adversity and manage emotions.	83%	1.00
8	- Promote the use of technological devices to maintain contact with relatives and close friends.	67%	1.00
	- Plan work tasks to adapt them to current regulations on prevention and to distribute workloads, with defined performance roles and access to human resource support.	50%	1.00
	- Apply COVID-19 infection prevention measures: social distance, disinfection, avoid closed spaces with poor ventilation.	33%	1.00
	- For people in situations of gender violence: remember to call 016 to request help with any situation (does not leave a telephone record). In a situation of danger, pharmacies activate the gender violence protocol when someone requests “Mask 19”.	67%	1.00
	- For people in a situation of danger or emergency: make sure they know that they can leave the home even during a period of confinement) to contact police, judicial, or other resources, without entailing a sanction.	50%	1.00
9	- Maintain sexual practices that avoid exposure to the risk of contagion when one of the partners has been in contact with infected people and presents any symptoms. There is no evidence of contagion by fluids.	67%	1.00
10	- Avoid focusing attention on the problem and look for alternative topics of conversation or activity.	17%	1.00
	- Know and perform relaxation techniques.	67%	1.00
11	- Seek information and support in making difficult decisions, avoiding making important decisions at times when emotions are very intense. If they cannot be postponed, make them based on decisions taken previously in uncertain periods.	50%	1.00

**Table 5 ijerph-19-08532-t005:** Not Approved Recommendations.

Functional Pattern	Recommendation	Contributing Participants %	CVI
1	- Monitor the cardiac system (heart rate, electrocardiogram for QT and torsade de pointes)	17%	0.40
	- Temporarily suspend mass vaccination campaigns (depending on the phase of the pandemic)	17%	0.20
2	- Supplement the diet with vitamin D in children and pregnant women and during breastfeeding. Systematic supplementation is not recommended in the general population.	17%	0.40
3	- Identify skin-type adverse reactions related to pharmacological treatment of COVID-19.	17%	0.40
4	- Use active video games.	17%	0.33
7	- Facilitate the development of a new body image.	17%	0.33
8	- Recommend surgical masks to the general population and health professionals not exposed to procedures that generate aerosols (PGA), and recommend the FFP2 type mask or higher only for health professionals before PGA. In times of material scarcity, reuse strategies can be considered.	17%	0.33
9	- Maintain an intrauterine copper device or *Levonorgestrel* for up to one year longer than the recommended time because it maintains safety.	17%	0.17
	- Consider menstrual hygiene a priority and ensure the follow-up of sexual and reproductive health needs.	17%	0.33
	- Avoid delaying the application of assisted reproduction treatment.	17%	0.33
	- Mark personal objects and communicate to sexual contacts the presence of COVID-19 symptoms and sexually transmitted diseases. The development of HIV to advanced stages of disease increases the risk of complications from COVID-19.	17%	0.33
11	- For the elderly: Identify emotions of fear of contagion and boredom due to lack of activity. Retired people show little uncertainty about the future and believe that it will return to normal.	17%	0.17

## Data Availability

The data that support the findings of this study are available on request from the corresponding author. The data are not publicly available due to privacy or ethical restrictions.

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
