# Peer review of "Care Recommendations for the Chronic Risk of COVID-19: Nursing Intervention for Behaviour Changes"

_ijerph, 2022, doi:10.3390/ijerph19148532_

Round 1
Reviewer 1 Report
Coexistence with the COVID-19 is a global health issue. This paper tried to provide recommendations regarding the coping strategies. However, there are several comments raised during my reading. Also, please use certificated English editing service.
1. COVID-19 was first emerged in December 2019.
2. The authors mentioned the meta-paradigmatic elements of care. It was unclear how these elements related to the manuscript.
3. Line 144, what 'summarized information' was sent to the participants in Phase I?
4. In Phase II, please provide reference for the validation. From my knowledge, the content validity index was counted by a 4-point Likert scale. Usually, 0.78 is used as a cutoff to indicate if the item is appropriate. I was not clear that CVI >=0.5 was being approved.
5. Please describe why the acquisition and validation were evaluated by the same group of six experts. Bias might be formed.
6. All the recommendations were from expert panel. Please describe why the guidances of COVID-19 care from international organizations were not included.
7. Please describe the characteristics of the participants. 'Experienced nurse' was not sufficient, such as, their specialty, working in hospital or community, ward of working, position as frontline nurse, head nurse, or nurse director, etc. The diversity of working place, position, and specialist experience could provide difference perspectives regarding the 'care' of COVID-19.
8. Physiotherapist was included in the expert panel. How about the other healthcare professionals? On the other hand, if the focus of this manuscript is 'care', why physiotherapist was included?
9. Line 240-242, it was unclear what nursing care model was based. If the guide was based on a care model, the authors should stated explicitly how the meta-paradigmatic elements (person, environment, and health) were adopted in the methods section.
10. Line 333-343, the authors tried to link the guide to clinical practice. However, the application of the guide was questioned. The guide was the opinions from 6 experts, which was not strong enough as other guide from international organizations. Further, it remains unclear how the recommendations from this guide were different or inferior to current guides or recommendations from international organizations, such as WHO.
Reviewer 2 Report
The topic in this manuscript is of importance, which is timely at the moment. However, below are several comments I need to emphasize.
Introduction
It is necessary to elaborate more on why the present research is based on a person-centred model and a nursing care model in the Introduction, as the authors described in the research design. It could be challenging for academic readers to understand the importance of the present research. The readers could be curious about the relevance between the person-centred model and the present main topic.
Materials and Methods
Please provide the conceptual framework for the present study.
2.1. Design
The person-centred theory understood by experts in their respective fields may be different. Readers are likely to wonder what kind of definition of person-centred theory was used and how it was explained to experts. Otherwise, did the authors have the operational definition? If so, please describe that in detail.
Page 3, lines 98-99
Please explain the Good Reporting of a Mixed Methods Study (GRAMMS) checklist in detail or provide the instrument as a supplement file.
Page 3, lines 108-109 and lines 132-133
The information about the expert group is crucial in the present study. The readers can validate the findings when the authors only provide information about the characteristics of experts in more detail. For instance, what expertise among experts, and how to recruit a group of experts? Moreover, how did the authors avoid conflicts of interest of experts?
Page 3, Table 1
What does it mean by “Date”?
Page 4, lines 171-174
How many experts participated in CVI, and how many times of rounds were performed to have a consensus among experts, and etc.? Moreover, have the authors evaluated the I-CVI-I and S-CVI/UA? The authors should present the values in diverse ways to improve the validation of the present findings. There have been several recommendations and theoretical evidence in previous literature. Thus, the authors can present them based on the evidence.
Results
Through the entire phase 1, 2 and 3, finding the relevance between person-centred theory and the present findings was challenging.
Discussion
Please reflect on the present findings based on the conceptual framework, named person-centred theory, in more detail.
Reviewer 3 Report
It was a pleasure reading the paper “Care recommendations for chronic risk of COVID-19: Nursing intervention for behaviour changes”. I believe it addresses an important topic and has the potential to strengthen the area of literature in improving COVID-19 care by standardizing nursing recommendations that include the multilayered complexities of the whole person and not just the clinical perspective.
The most evident strength is that the recommendations provided by this paper may be useful to pair with existing recommendations to cover each area of the patient's life (i.e., environment, mental, familial, etc.) rather than just the clinical treatment. However, the paper was sufficiently weak in describing what the authors actually did. The authors did not fully describe how they recruited the "experts" or what made these "experts" qualified to provide the recommendations. Further, the authors did not adequately describe how their methods were sufficient to methods used in creating existing international guidelines. For example, it seems like the experts were from one region and that the same experts were used for each phase of "validation" of the guidelines. This can create a bias by having the same "expert panel" review instead of asking for other experts to review. Further, it tends to limit external validity. The other weakness is that some English language corrections are needed so it was a bit challenging to read, especially in the introduction section.I have included specific recommendations below to strengthen this paper.
Abstract:
The authors should choose one tense to write in and stick with it. In some places the past tense is used, in others the present or future tense. It would be clearer if one tense was chosen.
Introduction:
Lines 36-38 can be restructured to read better. For example, it can be said “During the first months, health systems had to navigate a highly contagious, global illness with high morbidity and mortality, while having no prior evidence, leading to a restructuring of the available resources.”…or something of that nature.
Line 41: “two years later first cases were reported” should be changed to “two years after first cases were reported”
Line 43: “Nowadays” should be changed to something like “At present”, etc.
Line 45: Any reader who is not familiar with the term “SARS-CoV-2” may not realize that the authors are referring to COVID-19 here. It may be clearer to use this term in the first sentence of the Introduction section (line 34) when introducing the disease. For example: “The coronavirus disease, SARS-CoV-2, better known as COVID-19, emerged…” or something like that.
Line 47: “Nursing” does not need to be capitalized as a proper noun.
Line 59: “Nursing” does not need to be capitalized as a proper noun.
Line 87: “Nowadays” should be changed to something like “At present”, etc.
Line 87-88: This sentence is not clear as written. Please revise/reword for clearer understanding.
Line 90: “a care recommendations guide” can be reworded to “a guide for care recommendations” to be clearer.
Line 91: When the authors say “centred on the person”, what person are they referring to? The patient or the person making the care decisions? This can be made clearer.
Materials & Methods:
Line 96: “a care recommendations guide” can be reworded to “a guide for care recommendations” to be clearer.
Lines 96-97 can be rewritten as “Recommendations centred on the person and with a perspective based on nursing care models” is not a complete sentence.
Line 120: “used to registration in the…” does not read well. Authors could reword to “used for registration”.
Line 132: It may be helpful to detail what the authors mean by “intentional sampling”. It is unclear how the experts were recruited as the paper is currently written.
Results:
Table 2: I think there is a typo. The title for the first column should be “Functional pattern”.
Table 3: Same comment as table 2.
Table 4: Same comment as the other tables.
Line 231: “Finally, the recommendations explicitly related to 231 COVID-19” is not a complete sentence.
Table 5: Same comment as the other tables.
Discussion:
Line 285-288: I may have missed it, but I think it is important for the authors to reference what region they are referring to. Not every region follows Mediterranean diet or has adequate exposure to sunlight. This may be included as a limitation to the generalizability of this finding on regions outside of where the study was conducted.
Limitations:
See comment for Discussion section.
Conclusions:
Line 360: “Nursing” does not need to be capitalized as a proper noun.
Round 2
Reviewer 1 Report
The authors tried their best to address my comments in the revised manuscript. By reading the revised manuscript, several new concerns were raised.
1. It was unclear how the "Knowledge Model about Person Care" was. Please elaborate the model in terms of its meta-paradigm, i.e. person, environment, health, and nursing. Also, what essential components were included in the model and how they were interacted to improve an individual health?
2. Line 78-83, what 'model' was described in this section? The cited reference was not quite related to the "Knowledge Model about Person Care".
3. Orem's self-care deficit theory was described. It was confusing if the authors combined both "Knowledge Model about Person Care" and self-care deficit theory into this study.
4. Regarding the aim of the study, "the person who is trained to decision making by adapting or modifying their care behavior" was focused to against chronic risk of COVID-19. Please define who they were. From your explanation, the person should include physicians, nurses, and epidemiologists.
5. Line 119-120, please cite properly.
6. Line 158-165, characteristics of five experts were described. The aim of this study was to develop recommendations against chronic risk of COVID-19. The prevention was the main focus, but only two of the experts was not from hospital settings. The selection of experts was unclear. For example, the physicians from community could also make decision to adapt or modify the care behavior.
